# Effects and Economic Sustainability of Biochar Application on Corn Production in a Mediterranean Climate

**DOI:** 10.3390/molecules26113313

**Published:** 2021-05-31

**Authors:** Juan Luis Aguirre, María Teresa Martín, Sergio González, Manuel Peinado

**Affiliations:** 1Cátedra de Medio Ambiente, Department of Life Sciences, University of Alcalá, 28871 Madrid, Spain; maite.martin@edu.uah.es (M.T.M.); sergio.gonzalezegido@uah.es (S.G.); 2Environment and Bioproducts Group, Department of Life Sciences, University of Alcalá, 28871 Madrid, Spain; manuel.peinado@uah.es; 3Campus Universitario, Royal Botanical Garden of the University of Alcalá, 28801 Madrid, Spain

**Keywords:** biochar, corn, rentability, fast pyrolysis, soil analysis, economic analysis, pyrolysis

## Abstract

The effects of two types of biochar on corn production in the Mediterranean climate during the growing season were analyzed. The two types of biochar were obtained from pyrolysis of *Pinus pinaster*. B1 was fully pyrolyzed with 55.90% organic carbon, and B2 was medium pyrolyzed with 23.50% organic carbon. B1 and B2 were supplemented in the soil of 20 plots (1 m^2^) at a dose of 4 kg/m^2^. C1 and C2 (10 plots each) served as control plots. The plots were automatically irrigated and fertilizer was not applied. The B1-supplemented plots exhibited a significant 84.58% increase in dry corn production per square meter and a 93.16% increase in corn wet weight (*p* << 0.001). Corn production was no different between B2-supplemented, C1, and C2 plots (*p* > 0.01). The weight of cobs from B1-supplemented plots was 62.3%, which was significantly higher than that of cobs from C1 and C2 plots (*p* < 0.01). The grain weight increased significantly by 23% in B1-supplemented plots (*p* < 0.01) and there were no differences between B2-supplemented, C1, and C2 plots. At the end of the treatment, the soil of the B1-supplemented plots exhibited increased levels of sulfate, nitrate, magnesium, conductivity, and saturation percentage. Based on these results, the economic sustainability of this application in agriculture was studied at a standard price of €190 per ton of biochar. Amortization of this investment can be achieved in 5.52 years according to this cost. Considering the fertilizer cost savings of 50% and the water cost savings of 25%, the amortization can be achieved in 4.15 years. If the price of biochar could be reduced through the CO_2_ emission market at €30 per ton of non-emitted CO_2_, the amortization can be achieved in 2.80 years. Biochar markedly improves corn production in the Mediterranean climate. However, the amortization time must be further reduced, and enhanced production must be guaranteed over the years with long term field trials so that the product is marketable or other high value-added crops must be identified.

## 1. Introduction

Pyrolysis, a thermochemical degradation process performed in the absence of oxygen, is currently widely considered a viable option for waste treatment and generation of bioproducts [1]. Most forest residues generated in the field, as well as agricultural residues, are burned on-site as they do not have further applications. This results in the emission of greenhouse gases (GHGs) into the atmosphere and represents a major waste of raw material. One strategy for the management of these residues involves subjecting them to fast pyrolysis, which results in the production of the following four types of products: an aqueous fraction called wood vinegar; a heavy organic fraction called bio-bitumen; a light organic fraction known as bio-oil; a solid fraction called biochar. Additionally, fast pyrolysis results in the production of syngas, which mainly comprises of CH_4_, CO, H_2_, and CO_2_ that can feed the biomass heating process [2].

Pyrolysis of lignocellulosic residues can be a viable strategy if the resulting products are economically profitable. In particular, the economic benefits of bioproducts obtained through fast pyrolysis (wood vinegar, bio-oil, biochar, and bio-bitumen) should cover the cost of the pyrolysis treatment of these residues.

Wood vinegar has several applications, including herbicidal applications [3,4]. Several studies have demonstrated that wood vinegar can be used as a biostimulant and fertilizer [5,6,7] and that it enhances the beneficial effects of biochar [8]. Additionally, bio-oil has potential applications as a fuel [9], although the presence of oxygenated compounds can hinder its application. Bio-bitumen has been used as an additive to bitumen from fossil fuels, which reduces its carbon footprint and enhances its properties [10].

Biochar, which comprises of a porous carbonaceous structure with different functional groups, is an extensively researched product with numerous properties [11]. Biochar allows carbon fixation for at least several decades as it can function as a permanent carbon sink and hence can be used to tackle climate change. The emission of thousands of tons of CO_2_ into the atmosphere can be reduced if the emitted carbon from fires can be transformed into biochar [12]. Moreover, biochar exhibits adsorbent properties, which can improve soil properties [13,14,15] and promote water retention [16,17]. The supplementation of soil with biochar is reported to increase the availability of basic plant nutrients [18].

Various studies have focused on the effect of biochar supplementation on soil fertility and the productivity of crops, such as corn [19,20,21,22], grapevine [23], wheat [24], rice [25,26], barley [27], cotton [28], and potato [29]. Most studies have demonstrated the efficacy of biochar supplementation on crop productivity, while some studies have reported limited or no efficacy of biochar addition in temperate regions where soil fertility is sufficiently high [30,31]. In a review on the barriers to the start-up of biochar application in agriculture, Guo et al. 2016 refers to the gap between research and application, immaturity of economical production technologies, lack of application-specific quality standards and management programs [11]. 

This study aimed to determine the effect of biochar supplementation on corn production in a Mediterranean region in a single growing season. Two types of biochar from *Pinus pinaster* were tested. Biochar is produced in an industrial reactor of NEOLIQUID technology with internal agitation and external heating. The reactor consists of a screw type reactor with a diameter of 250 mm and 6 m length. Biochar B1 comes from a complete pyrolysis process with a residence time of 30 min. However, biochar B2 is produced at a residence time of 15 min. 

Additionally, the period for which enhanced productivity should persist for the product to be profitable for a farmer is discussed. Corn was selected in this study as it is one of the most productive and cultivated plants worldwide and improvements in corn production can promote the use of biochar. Most studies examining the effect of biochar on corn plantations were performed in tropical or template climates with limited studies performed in Mediterranean climates where the climatic limitation of rainfall makes it a very irrigated crop. Corn is also one of the most profitable crops in this type of irrigated soil, hence minimal improvements in corn production can benefit the farmers.

## 2. Results

### 2.1. Biochar Composition

Biochar 1 (B1) and biochar 2 (B2) exhibited markedly different compositions. B1 exhibited higher significant levels of organic carbon, organic matter, total humic extract, carbon/nitrogen (C/N) ratio, and fulvic acid but decreased levels of humic acid and moisture (Table 1). The other variables were not different between the two biochar samples. These findings along with the morphological characteristics of the two types of biochar suggest that B1 was completely pyrolyzed, while B2 underwent partial pyrolysis (non-pyrolyzed wood chips were appreciated).

### 2.2. Soil Analysis

The analysis of soil properties indicated that soil fertility was high at the time of planting. All soils were deep soils, fertile, and derived from an alluvial terrace with the following characteristics: organic matter content, >3% in most cases; well structured; balanced proportions of sand, silt, and clay; pH, 7.6–7.8. The properties of the soils were not significantly different before planting (Table 2).

No differences were observed in most of the fertility-related variables, such as total nitrogen, potassium, and assimilable phosphorus analyzed after harvesting. The conductivity and carbon to nitrogen (C/N) ratio of the soil increased after harvesting. The levels of sulfate, magnesium, and saturation percentage were significantly higher in the soils supplemented with biochar. Soils supplemented with B1 retained higher contents of nitrates than the other soils after harvesting.

### 2.3. Harvest Data

#### 2.3.1. Cob Morphological Data

The cob insertion height (Fisher’s least significant difference (LSD) test; *p* > 0.05 in all cases) was not different between different treatment groups. The cob length in the plots with biochar (B1 and B2) was higher than that in the control plots (C1 and C2) (Fisher LSD; *p* < 0.01). Additionally, the length of cobs was significantly different between C1 and C2 plots (Fisher LSD; *p* < 0.05) but not between the B1-supplemented and B2-supplemented plots (Table 3). The perimeter of cobs in the B1-supplemented plots was significantly higher than that in the B2-supplemented, C1, and C2 plots (Fisher LSD *p* < 0.01). The cob diameter was significantly different between B1-supplemented and other plots (Fisher LSD; *p* < 0.01), and between B2-supplemented and C2-supplemented plots (Fisher LSD; *p* < 0.05). The percentage of cobs with few grains (assumed to be due to the lack of pollination or incomplete pollination) was not different between treatment groups (C1 plots, 25.75%; C2 plots, 21.01%; B1-supplemented plots, 20%; B2-supplemented plots, 28.30%).

#### 2.3.2. Fresh Cob Production Data

The weight of cobs in B1-supplemented plots was significantly higher than that of cobs in other plots (Fisher LSD; *p* < 0.01) (Figure 1). However, the weight of cobs was not different between the B2-supplemented, C1, and C2 plots (Fisher LSD; *p* > 0.05) (Table 4).

The number of cobs per plant (including plants with two cobs) in B1-supplemented plots was significantly higher than those in the other plots (Fisher LSD; *p* < 0.01). However, the number of cobs per plant was not different between the B2-supplemented, C1, and C2 plots. Additionally, the number of cobs per plant was significantly different between the C1 and C2 plots (Fisher LSD; *p* < 0.05). In B1-supplemented, B2-supplemented, C1, and C2, 17.6%, 1.9%, 26%, and 0% of the plants had two cobs, respectively.

The weights of 100 corn grains in B1-supplemented plots were significantly higher than those in other plots (Fisher LSD; *p* < 0.01). Meanwhile, the weights of 100 corn grains in C1 plots were significantly higher than those in C2 and B2-supplemented plots (Fisher LSD; *p* < 0.01). The weights of 100 corn grains were not different between B2-supplemented and C2 plots (Fisher LSD; *p* > 0.05).

B1-supplemented plots produced significantly more corn than the other plots (Fisher LSD; *p* < 0.01). However, there was no difference between B2-supplemented, C1, and C2 plots (Fisher LSD *p* > 0.05). The number of plants in C1, C2, B1-supplemented, and B2-supplemented plots was 52, 57, 68, and 52, respectively.

#### 2.3.3. Dried Cob Production Data

The moisture content in the whole fresh cobs was 26.33% ± 4.51 with no differences between treatment groups (Fisher LSD; *p* > 0.05). The mean moisture content of the corn grains was 18.82 ± 4.34. Additionally, the mean moisture content of corn grains from the B1-supplemented (19.85 ± 1.18) and B2-supplemented (22.13 ± 2.62) plots was significantly higher than that of the corn grains from the C1 (14.77 ± 1.73) and C2 (12.45 ± 2.12) plots. To estimate the dry grain production, a linear regression curve of the fresh cob weight against the grain weight in each plot was generated, and the grain weight was estimated for all cobs (Table 5). The R^2^ and *p* values in all cases were >0.99 and <0.01, respectively. The moisture correction for each treatment was applied to obtain the result of the dry corn grain.

The weight of dry grains per cob in B1-supplemented and B2-supplemented plots was significantly higher than that in C1 and C2 plots. Additionally, the dry grain weight per cob was significantly different between B1-supplemented and B2-supplemented plots (Fisher LSD; *p* < 0.01) but not between C1 and C2 plots (Fisher LSD; *p* > 0.05). The dry grain weight per plant (including the plants with two cobs) in the B1-supplemented plots was significantly higher than that in the other plots (Fisher LSD < 0.01). Additionally, the dry grain weight per plant in the C1 plot was significantly different from that in the other plots (Fisher LSD; *p* < 0.01) but was not different between C2 and B2-supplemented plots (Fisher LSD; *p* = 0.89).

The dry grain weight per square meter was significantly different between B1-supplemented and other plots (Fisher LSD; *p* < 0.01) but was not different between B2-supplemented, C1, and C2 plots (Fisher LSD; *p* > 0.05) (Figure 2).

## 3. Discussion

The effect of biochar on the growth of several plant species has been studied in tropical climates. For example, the beneficial effect of biochar on the growth of rice has been previously reported [26]. Several studies examining the effect of biochar on corn production in Africa, the USA, and Asia have reported that the efficacy of biochar to increase the production of corn ranges from 80–100% at high concentrations, to no significant results in some cases [32,33]. At low doses, the effect of biochar on plant growth is mitigated. For example, Borchar et al. reported that the supplementation of biochar at doses ranging from 15 g to 100 g/kg did not result in beneficial effects on corn production [34].

In warm climates, the effect of biochar supplementation on plant growth ranged from highly ineffective [35] to highly effective [36]. The effect of biochar on plant species in the Mediterranean climate was examined in control chambers with limited field studies. For example, Baronti et al. reported that corn production increased by 6% in Italy upon supplementation with biochar at a dose of 10 t/ha of soil. The production of perennial grasses increased by 120% upon supplementation with biochar at a dose of 60 t/ha of soil [24]. However, biochar supplementation did not markedly affect the crop production in temperate areas, such as New York with a rainfall of more than 900 L although the retention of added N increased [30]. The production of wheat (*Triticum durum* L.), sunflower (*Helianthus annus*), or annual species, such as *Lupinus* increased in the Mediterranean climate upon supplementation of biochar from olive wood [37,38,39].

The results of this study demonstrated that in contrast to B2, B1 enhanced cob weight, grain weight, and cob production. The dry grain weight per square meter in B1-supplemented plots increased by 84.58% when compared with that in C1 and C2 plots (63.37% higher when compared to the best results). The fresh weight in B1-supplemented plots increased by 93.69% when compared with that in C1 and C2 plots (73.18% higher when compared to the best plots). This indicated that the application of B1 (but not B2) enhanced corn production.

The variability in biochar definition can account for the lack of homogeneity of the results reported in various studies. For example, Jones et al. reported that biochar did not affect corn growth upon supplementation with amounts similar to those used in this study [40]. However, the authors examined vegetation data but did not analyze harvest data. In a recent review, Ippolito et al. analyzed data from multiple experiments and concluded that temperature and origin of raw material may explain the variability in the effects of biochar [41]. Rajkovich et al. studied the effect of the starting raw materials and the biochar pyrolysis temperature on corn production. The total biomass production was similar upon treatment with 0.2%, 0.5%, and 2% but significantly decreased upon treatment with 7% biochar. However, the authors reported significant differences in the effects of different types of biochar. The type of raw material had an eight times higher effect on corn biomass production than pyrolysis temperature [21]. Butnan et al. examined the addition of two types of biochar obtained by different processes at different temperatures (350 and 800 °C). Biochar obtained at 350 °C was highly effective in improving soil properties and corn growth [42]. In the same line, Guo 2020 points out three main factors to improve the biochar application effectiveness: right biochar source, right application rate, and right placement in soil [43].

Spokas et al. suggested that products with limited commonalities are considered equivalent [32]. For example, Manyá recommends that it is important to describe the biochar composition and main variables when its effects on production are reported [44]. The results of this study confirm that two types of biochar obtained from pine wood with different compositions exhibited differential effects.

In addition to its chemical and biological properties, the structural properties of biochar contribute to its improved performance in sandy soils or areas with water deficit as its porous structure can increase the water retention capacity [14,45] and consequently enhance the water available to plants [46,47]. In this study, B1 (but not B2) directly and markedly improved corn production.

The soil analysis revealed that the application of biochar did not affect most soil nutrients, such as nitrogen, phosphorus, and potassium. However, biochar affected some soil properties, such as conductivity, sulfate, magnesium, or saturation percentage. Recently, Olmo et al. reported that biochar supplementation decreased the bulk density and compaction of the soil and increased its water retention capacity. This is consistent with the results of this study as corn is cultivated in a Mediterranean climate where water availability is a determining factor for its growth [39]. Other authors have reported that biochar supplementation increases the conductivity of soil [48,49]. Furthermore, biochar enhances soil nutrient retention. The findings of this study indicated that B1 supplementation increases the Mg content. The increased availability of Ca and Mg in the soil promotes crop production [18,50].

### Profitability of Biochar Application

Both short-term and medium-term profitability must be ensured for the successful application of biochar by farmers or administrations. Economic profitability is an essential factor for the application of biochar as a carbon sink by farmers. Filiberto and Gaunt developed an economic model in 2013 to determine the economic value of biochar application. They indicated that the cost of fixing one ton of environmental CO_2_ should be $87.5 to justify the costs [51]. Therefore, the agronomic value of biochar should cover the difference between this cost and the real cost of fixing one ton of CO_2_.

In the Mediterranean region, a ton of corn with 14% moisture content is approximately €185 [52]. One hectare of land in the Mediterranean area can generate approximately 12 tons [53]. Therefore, a 63% improvement in dry grain production (the improvement rate relative to the best improvement observed in the untreated plot) can result in the yield of an additional 7.44 t. If the price for one ton of corn is €185, the input increase in one hectare is €1376.4, considering that the harvest costs are similar regardless of the harvest amount. (Table 6). This amount would be the maximum cost that a farmer could afford in a year without losing profits. The findings of this study indicate that biochar must be incorporated at a dose of 4 kg/m^2^ (40 t/ha).

Based on the estimated cost of approximately €190 per ton of biochar, which is the medium price of this product on some websites in Spain, (a), the cost of its application at 4 kg/m^2^ would be approximately €7600. However, other authors have estimated a cost of approximately €6000 [51]. Therefore, biochar should improve corn production for at least 5.52 years to make its purchase profitable assuming that it can be implemented at a low cost with the machinery of the farmers. Similarly, Dokoohaki et al. indicated that the application of 5–15 t of biochar per hectare is efficient in areas of the USA with poor soils for corn production with increased production maintained for 5 to 10 years. However, these applications are not efficient for other studied crops (soybeans and wheat) or high-quality soils [54].

One option to reduce the costs is to include the subsidized price of one ton of CO_2_ that is no longer emitted in the income of farmers or by reducing the cost of biochar (for economic purposes). According to the data provided by Filiberto and Gaunt in 2013, one ton of biochar reduces the emission of 2.06 tons of CO_2_ [51]. The price of CO_2_ in 2021 has increased to 50 € per ton, but a price of €30 per ton is applied because it may be more realistic in the long term. Therefore, the application of biochar provides €2472/ha in emission savings if 40 tons are used, which would reduce the cost for farmers by approximately one-third.

Several studies have reported the reduction of expenses with fertilization and irrigation. However, it is difficult to calculate costs with such data. If biochar could be used to avoid fertilization, the cost can be reduced by 50% for the unfertilized plots. Considering an estimated cost of approximately €507.95 per hectare of corn maize [55], the estimated saving is €253.97. Moreover, the cost of water is estimated to be 38% of the total cost in the Mediterranean irrigated areas [56], which is approximately €700–800 per hectare. As water prices are high, savings in water efficiency and fertilization can enable the profitable application of biochar. Considering an annual saving of 25% of water (€200) and a similar saving for fertilization, this could generate an annual improvement of more than €453.97 per hectare, which can amortize the investment made by the farmer for the incorporation of biochar at €190/t in three years (Table 7).

The results obtained in this study for corn must be replicated and maintained for at least 4 years for the use of biochar to be profitable. Some studies estimate productivity improvements for 5 to 10 years, while others consider up to 30 years in soil improvement [57]. Thus, while short-term improvements are important for the farmer and it is essential to obtain short-term results in this type of business, more long-term trials should be carried out. Additionally, the incorporation of the value of non-emitted CO_2_ through CO_2_ bonds or compensation for companies will recover the investment in approximately 3 years in the case of corn, which is reasonable if constant improvements are expected for about 10 years.

These findings indicate that the application of biochar can be profitable in high value-added crops for which production increases have a high economic value. However, the use of biochar is discouraged in low value-added products, such as wheat and barley.

## 4. Materials and Methods

### 4.1. Biochar

Two types of biochar from pine wood chips with different characteristics were used. Pyrolysis was performed in a semi-industrial continuous feed pyrolysis plant, which produces biochar, wood vinegar, and bio-oil, with a syngas circulation system to feed the biomass heating. This prototype was developed as part of the EU Lignobiolife project, CCM/ES000051.

### 4.2. Field Experiments

The field experiments were performed using the facilities of the Royal Botanic Gardens, University of Alcalá (Madrid, Spain). Four experimental fields were divided into 40 randomly arranged plots each with an area of 1 m^2^. Biochar B1 and biochar B2 were supplemented in 20 plots (10 plots/supplementation). In C1 and C2 (10 plots/each) plots, no fertilizer or compost was used. B1-supplemented and C1 plots were separated from B2-supplemented and C2 plots by approximately 10 m. Biochar (4 kg/m^2^) was added in December 2019 and mixed with the soil using a power tiller.

Each plot had two planting rows with a spacing of 50 cm between rows and 30 cm between plants. In total, 272 plants emerged and developed. The plots were protected from rabbits and rodents with a metal mesh. For protection against birds, a semi-rigid plastic mesh was placed in an arch with an enclosure height of 2 m.

A drip irrigation system regulated with a solenoid valve and TBOSS II programmer was installed in all plots.

The corn was harvested in the first week of September when the cobs had already developed and the leaves were starting to turn yellow. The number of plants in C1, C2, B1-supplemented, and B2-supplemented plots was 52, 57, 68, and 52 plants, respectively. The cob for each plant was identified. The height of the insertion point of the cob in the plant and the cob number was recorded. The diameter, perimeter, length, weight, and number of rows in the center were measured for each cob. Additionally, the fully developed cobs covered with grains were recorded. The largest and the smallest cobs were collected from each plot (*n* = 40), and the grains and cobs were weighed separately. Additionally, the weight of 30 grains was recorded. Subsequently, the weight of grain in each cob was estimated using Pearson’s linear regression analysis.

Furthermore, 80 cobs were selected (20 cobs per plot) and dried to estimate the difference between fresh weight and dry weight of both whole cobs and grains. A linear regression curve was generated to estimate the corn dry weight for all cobs.

### 4.3. Soil Analysis

In total, eight soil samples were collected before the application of biochar in the plots before planting and eight soil samples were collected after the application and harvest. In total, 16 samples were analyzed. The superficial 25 cm of soil was selected to analyze the pH, conductivity at 20 °C, sodium adsorption ratio, chloride, nitrate, sulfate, phosphate, carbonate, bicarbonate, saturation moisture content, sodium, potassium, calcium, magnesium, cation ratios and exchange (sodium, potassium, calcium, magnesium, cation exchange capacity (CEC)), total nitrogen, assimilable potassium, assimilable phosphorus, easily assimilable organic matter, C/N ratio, and texture (sand, silt, and clay) of each sample. The analyses were performed under controlled laboratory conditions (Tentamus Company, Las Rozas, Madrid, Spain).

### 4.4. Biochar Analysis

The biochar supplied by Neoliquid was weighed and measured. The digestion, gravimetry, moisture content at 105 °C (UW), electrical conductivity at 25 °C, organic matter, organic carbon, carbon/nitrogen ratio, total humic extract, humic acid and fulvic acid, in each biochar were analyzed at Eurofins Agroambiental S (Sidamon, Lleida, Spain).

### 4.5. Statistical Analysis

The data were analyzed using analysis of variance, followed by Fisher’s test. Linear regression analyses were performed to estimate grain weight per cob. All statistical analyses were performed using Statplus 7.1. (AnalystSoft Inc., Walnut, CA, USA).

### 4.6. Economic Analysis

Biochar prices were obtained by reviewing different websites in Spain and Europe [58,59,60]. The price per ton of corn and the costs of its cultivation, fertilizers and water were taken from the average values in 2020 in Spain from agricultural organizations (54, 57). The cost per ton of CO_2_ was the average for 2020 and early 2021, averaging 30 €/t, although in 2021 it has reached 50 €/t.

## 5. Conclusions

The results of this study demonstrated that the application of biochar B1 increased corn production by 63.4–84% based on the dry grain weight in the first year. However, biochar B2, which was partially pyrolyzed, did not significantly affect corn production. The soil supplemented with B1 exhibited enhanced levels of sulfate, magnesium, and saturation percentage in addition to increased retention of nitrates after planting. It is estimated that it would take approximately 4 years to amortize the investment, with an estimated price of €190 per ton of biochar and also considering a 50% reduction of the fertilization costs and a 25% reduction of the irrigation costs with the maintenance of corn production over the years. If the CO_2_ emission is reduced by 2.06 tons using one ton of biochar and sold to the emissions market, the amortization could be reduced to approximately 3 years. These data demonstrated a marked improvement in crop production with high-quality biochar and provided a strategy for the utilization of this type of material in high value-added crops. However, the economic profitability will be low in low value-added crops.

## Figures and Tables

**Figure 1 molecules-26-03313-f001:**
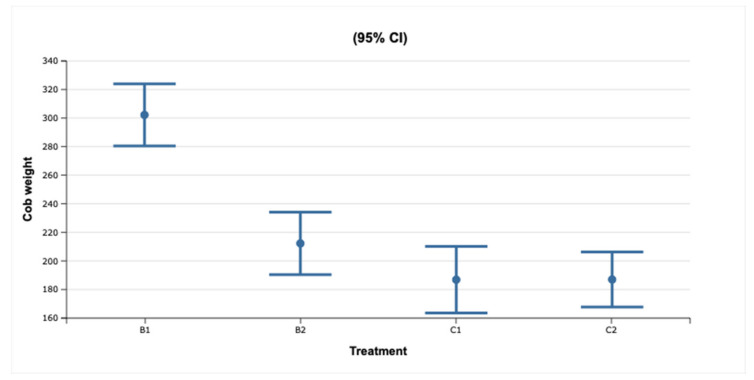
Average fresh cob weight in different treatment groups. Control: plots without biochar; B1: plots supplemented with biochar B1; B2: plots supplemented with biochar B2.

**Figure 2 molecules-26-03313-f002:**
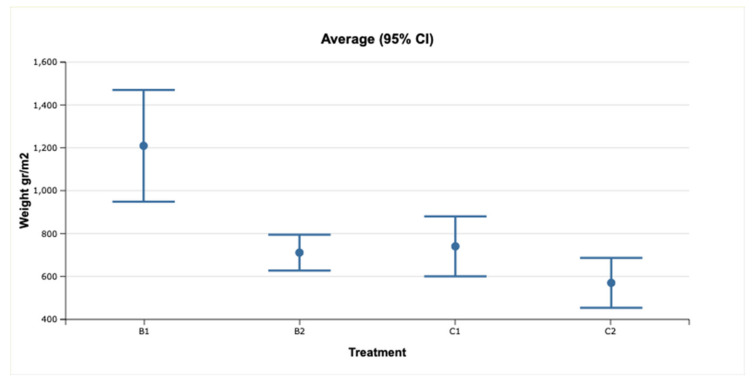
Dry grain weight per square meter in different treatments (*n* = 10). Mean ± standard deviation. Control: plots without biochar; B1: plots supplemented with biochar B1; B2: plots supplemented with biochar B2.

**Table 1 molecules-26-03313-t001:** Basic properties of the two types of biochar used in the experiments. B1: Biochar type 1, B2: Biochar type 2.

	B1	B2
Dry matter	98.70%	81.70%
Moisture	1.33%	18.30%
pH	8.6	7.9
Electrical conductivity (25 °C)	0.508 dS/m	7.54 dS/m
Organic matter (550 °C)	97.7%	49.5%
Organic carbon	55.90%	23.50%
C/N ratio	233.99	33.05
Humic acid	0.9%	4.9%
Total humic extract	20.3%	8.3%
Fulvic acid	19.40%	3.40%

**Table 2 molecules-26-03313-t002:** Number of macronutrients and soil characteristics in soil before and after planting. Control: plots without biochar; B1: plots supplemented with biochar B1; B2: plots supplemented with biochar B2.

	Before Planting (n = 8)	Control (n = 4)	B1 (n = 2)	B2 (n = 2)
pH	7.8 ± 0.06	7. 9 ± 0.04	8.01 ± 0.0	7.85 ± 0.0
Conductivity (µs/cm)	264 ± 49 *	293 ± 28 *	522 ± 12 *	346 ± 14 *
Total N (%)	0.17 ± 0.05	0.15 ± 0.02	0.15 ± 0.01	0.21 ± 0.06
K assimilable (mg/kg)	397 ± 246	291 ± 82	453 ± 22	444 ± 99
P assimilable (mg/kg)	50.8 ± 31.8	30.54 ± 6.98	38.13 ± 11.7	37.47 ± 25.5
Organic matter (%)	3.5 ± 1.01	3.25 ± 0.23	4.32 ± 0.68	5.46 ± 1.39
C/N	12.35 ± 1.1 *	12.4 ± 0.88 *	16.68 ± 1.08 *	15.10 ± 0.22 *
Sulfate (mg/L)	8.13 ± 3.4 *	38.25 ± 6.7 *	142.5 ± 6.36 *	53 ± 5.8 *
Magnesium (mg/L)	6.43 ± 2.4 *	9.5 ± 1.9	14.5 ± 0.7 *	10.5 ± 0.7
Saturation percentage (%)	29.1 ± 1.3 *	29.9 ± 0.5 *	33.64 ± 1.9 *	33.4 ± 1.4 *
Phosphate (mg/L)	5.55 ± 2.76	2.7 ± 0.4	2.5 ± 0	3.75 ± 1.7
Nitrate (mg/L)	28 ± 2.83 *	9 ± 2.83	23 ± 5.6 *	4.5 ± 0.71

* Significant differences analyzed using analysis of variance, followed by Fisher’s test (*p* < 0.01).

**Table 3 molecules-26-03313-t003:** Cob morphological data based on treatment. Data are expressed as mean ± standard deviation. Control: plots without biochar; B1: plots supplemented with biochar B1; B2: plots supplemented with biochar B2.

Treatment/Cob	Cob Insertion Height (cm)	Length (mm)	Perimeter (mm)	Diameter (mm)
B1	81.88 ± 8.86 *	217.87 ± 24.66 **	155.18 ± 29.31 *	48.79 ± 6.53 *
B2	80.55 ± 8.84	229.89 ± 75.53 **	144.88 ± 26.39	44.52 ± 4.08 *
C1	82.81 ± 13.30	197.21 ± 27.24 *	140.33 ± 9.93	42.05 ± 6.01
C2	80.78 ± 11.13	176.49 ± 27.24 *	140.29 ± 18.82	43.96 ± 3.38

Fisher significance test *: *p* < 0.01; ** *p* < 0.001.

**Table 4 molecules-26-03313-t004:** Cob production and weight data based on treatment. Data are represented as mean ± standard deviation. Control: plots without biochar; B1: plots supplemented with biochar B1; B2: plots supplemented with biochar B2.

Treatments	Fresh Cob Weight (g/Plant)	Fresh Cob Weight (g/Cob)	Weight of 100 Grains (g)	Fresh Weight Productivity (g/m^2^)
B1 (68 plants)	307.28 ± 138.97 **	302.2 ± 84.94 **	43.64 ± 5.97 **	2028.05 ± 542.87 **
B2 (52 plants)	202.85 ± 79.20	210 ± 16	36.43 ± 4.53	1117.63 ± 189.7 *
C1 (52 plants)	231.07 ± 135.22 *	186.5 ± 84.87	39.46 ± 4.69	1171.39 ± 290.32
C2 (57 plants)	163.28 ± 74.77 *	187.00 ± 64.0 *	31.36 ± 5.40	923.43 ± 228

Fisher significance test *: *p* < 0.01; ** *p* < 0.001.

**Table 5 molecules-26-03313-t005:** Production data, dry weight and yield per square meter. Data are represented as mean ± standard deviation. Control: plots without biochar; B1: plots supplemented with biochar B1; B2: plots supplemented with biochar B2.

Treatment	Equation (Cob Weight and Grain Weight)	Dried Grain Production Per Cob (g)	Dried Grain Production Per Plant (g)	Dried Grain Productivity (g/m^2^)
B1	Grain weight = −17.05 + 0.82 * Cob weight (R^2^: 0.99)	152.97 ± 71.88 **	177.83 ± 86.52 **	1209.24 ± 364.57 **
B2	Grain weight = −3.39 + 0.79 * Cob weight (R^2^: 0.99)	131.69 ± 49.32 *	136.76 ± 56.62	711.14 ± 116.8 *
C1	Grain weight = −21.41 + 0.85 * Cob weight (R^2^: 0.99)	110.74 ± 60.11	138.79 ± 89.98	740.32 ± 195.20
C2	Grain weight = −16.35 + 0.82 * Cob weight (R^2^: 0.99)	100.05 ± 52.25	100.05 ± 52.25 **	570.31 ± 162.54

Fisher significance test *: *p* < 0.01; ** *p* < 0.001.

**Table 6 molecules-26-03313-t006:** Main variables in the application of biochar in corn.

Biochar/ha	Cost of a Ton of Dry Corn	Average Production	Improvement/ha	Increasing Input
40 t/ha	€185	12 t/ha	7.44 t	€1376.4/ha

**Table 7 molecules-26-03313-t007:** Potential savings with the use of biochar.

CO_2_ Value	Savings in Fertilization	Irrigation Savings	Biochar Cost €190/t	Crop Improvement (Corn Value)	Time to Amortize (without CO_2_)	Time to Amortize (with CO_2_)
€2472/ha	€253	€200	€7600	€1376.4	4.15 years	2.80 year

## Data Availability

Data can be requested from the author J.L.A.

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
