# Peer review of "Effects and Economic Sustainability of Biochar Application on Corn Production in a Mediterranean Climate"

_molecules, 2021, doi:10.3390/molecules26113313_

Round 1

Reviewer 1 Report

The manuscript entitled “Effects and Economic Profitability of Biochar Application on Corn Production in a Mediterranean Climate” aims to determine the effect of biochar supplementation on corn production in a Mediterranean region in a single growing season. The research is conducted very well, and the manuscript is written in a scientifically sound manner. Moreover, the presented results are of great importance for the environment. It would be interesting to check the influence of biochars on pesticides from soil. Also, the exact method for two biochar syntheses should be given.

Author Response

Thank you very much for your words. Unfortunately we don't analyze pesticides in this study, because the trail were made in Royal Botanical garden of University of  Alcalá that don't use pesticides in their areas. 

We describe the exact method for biochar syntheses, and the difference between B1 and B2 Biochar. 

Reviewer 2 Report

I am not sure if the theme of “economic profitability” is fitting in the scope of the journal. Maybe “sustainability” is a best outlet for this type of manuscripts.

L12 and in the main text. Authors should provide the information on the raw materials of the two types of biochar B1 and B2. The type of Biochar source is a very critical, if not most, factor to determine a biochar’s chemical properties and functions.

L16 and throughout the manuscript. Please reserve the term “significantly” and alike to present statistically treated data/observations, and with the probability (p values) given.

LL20-21. Can you give some values or percentage ranges of the increased levels?

L22. Clarify if “€190 per ton of biochar” is for B1 or both B1 and B2.

L32. The introduction must be rewritten. In the current version, the first five paragraphs of introduction reads distracting, not focused on your project theme. For example, why did you talk about “…have applications in the chemical recycling of the components (especially plastics or tires [2]) or the generation of commercially important bioproducts [3]”. The topic is totally irrelevant of your work.

The Introduction can be started from the sixth paragraph (L67) which reads more relevant. Expanding the last three paragraphs for the new introduction. May cite a biochar book at with the beginning sentence (e. g., Agricultural and Environmental Applications of Biochar: Advances and Barriers. SSSA Special Publication 63. Soil Science Society of America. Madison, Wisconsin).  

Also need to elaborate somehow on the effects of source materials on the biochar properties to justify you selected two types of the biochars in this work. Some chapters in the biochar book could be helpful.

Table and figure captions should be self-explanatory. Authors should give the definition of C1, C2, B1 and B2 so that readers can understand their meanings without consulting the main text.

“Data are represented as mean ± standard deviation” is fine, but also give the replcates (n=?).

Figure 1 caption is not consistent with Figure 2

LL289-297, conclusion and abstract. Authors should be cautious on the benefit calculation extended to multiple years down the road as you reported one-time (year) data. At least, you should mention multiple year data is needed in future work to confirm the results…. Refer to challenges and barriers listed in the last chapter of the biochar book. “long-term field trials are needed…”

Materials and methods should list from where and/or how those economic data (prices) were obtained.

Author Response

I am not sure if the theme of “economic profitability” is fitting in the scope of the journal. Maybe “sustainability” is a best outlet for this type of manuscripts.

Taking into account this consideration, “economic profitability” has been replaced by“economic sustainability”. Our intention is to indicate in the title that an economic analysis is made in the paper.

L12 and in the main text. Authors should provide the information on the raw materials of the two types of biochar B1 and B2. The type of Biochar source is a very critical, if not most, factor to determine a biochar’s chemical properties and functions.

We agree. Both of them are from Pinus pinaster wood.

L16 and throughout the manuscript. Please reserve the term “significantly” and alike to present statistically treated data/observations, and with the probability (p values) given.

We do, we erased “significantly” in all cases with no statistical differences. We give the p values in the abstract too.

LL20-21. Can you give some values or percentage ranges of the increased levels?

In this case we prefer refers to the table because each variable has different increased value. As this paper would be published as open access all interested people can check it in the paper.

L22. Clarify if “€190 per ton of biochar” is for B1 or both B1 and B2.

190 € is the price of buying a ton of Biochar from some companies in Spain. We have now included the references of some websites in the manuscript.

L32. The introduction must be rewritten. In the current version, the first five paragraphs of introduction reads distracting, not focused on your project theme. For example, why did you talk about “…have applications in the chemical recycling of the components (especially plastics or tires [2]) or the generation of commercially important bioproducts [3]”. The topic is totally irrelevant of your work.

The Introduction can be started from the sixth paragraph (L67) which reads more relevant. Expanding the last three paragraphs for the new introduction. May cite a biochar book at with the beginning sentence (e. g., Agricultural and Environmental Applications of Biochar: Advances and Barriers. SSSA Special Publication 63. Soil Science Society of America. Madison, Wisconsin).  

Introduction has been reviewed. First paragraphs have been reduced and erased. However, we maintain the references to the bioproducts produced in the pyrolysis process. The profitability of this technology is linked to the use, and sell, of biochar, wood vinegar, bio-oil and bio-betumen. We revised and cite this interesting book and other new references.

Also need to elaborate somehow on the effects of source materials on the biochar properties to justify you selected two types of the biochars in this work. Some chapters in the biochar book could be helpful.

 We explain the process to obtain biochar, with the data of pyrolysis process and the differences between B1 and B2 Biochar. We revised some papers of that book.

Table and figure captions should be self-explanatory. Authors should give the definition of C1, C2, B1 and B2 so that readers can understand their meanings without consulting the main text.

Definition of C1, C2, B1 and B2 has been included in figures and tables.

“Data are represented as mean ± standard deviation” is fine, but also give the replcates (n=?).

n value has been incorporated in the most interesting data.

Figure 1 caption is not consistent with Figure 2

Figure captions have been reviewed and corrected.

LL289-297, conclusion and abstract. Authors should be cautious on the benefit calculation extended to multiple years down the road as you reported one-time (year) data. At least, you should mention multiple year data is needed in future work to confirm the results…. Refer to challenges and barriers listed in the last chapter of the biochar book. “long-term field trials are needed…”

We do it. We refer to the importance of long term studies although some of them have been refer in the manuscript.

Materials and methods should list from where and/or how those economic data (prices) were obtained.

Section 4.6. Economical Analysis has been included. In this section it is explained where biochar price has been obtained. Data related to corn and production costs is referred to Spain averages in the last years obtained from official sources. Data of carbon price changes all months but a conservative value of 30 € tn was used in the analysis, although when we write these lines have a record of 51 €/tn. This data is very important for the economic analysis but we doubt that will be stable in time. The cost of biochar is obtained from some Spanish and Europe companies.

Round 2

Reviewer 2 Report

The revision is acceptable